# Impact of Delayed Admission on Treatment Modality and Outcomes of Aneurysmal Subarachnoid Hemorrhage: A Prefecture-Wide, Multicenter Japanese Study

**DOI:** 10.3390/jcm14103537

**Published:** 2025-05-18

**Authors:** Yuma Hosokawa, Hitoshi Fukuda, Yuki Hyohdoh, Takako Kawamura, Ken Shinno, Yongran Yanase, Masaki Yokodani, Yu Hoashi, Akihito Moriki, Koji Bando, Nobuhisa Matsushita, Fumihiro Hamada, Yu Kawanishi, Yusuke Ueba, Naoki Fukui, Noritaka Masahira, Yo Nishimoto, Tetsuya Ueba

**Affiliations:** 1Department of Neurosurgery, Kochi Medical School, Kochi University, Nankoku 783-8505, Japan; uma55yh@gmail.com (Y.H.); hamada-fumihiro@kochi-u.ac.jp (F.H.); ykawanishi@kochi-u.ac.jp (Y.K.); yueba@kochi-u.ac.jp (Y.U.); naofukui@kochi-u.ac.jp (N.F.); tueba@kochi-u.ac.jp (T.U.); 2Center of Medical Information Science, Kochi University, Kochi 783-8505, Japan; yuki_hyohdoh@kochi-u.ac.jp; 3Department of Neurosurgery, Aki General Hospital, Aki 784-0027, Japan; pukutaka777@gmail.com; 4Department of Neurosurgery, Chikamori Hospital, Kochi 780-0052, Japan; k.shinno.1017@gmail.com (K.S.); yoh.nishimoto@gmail.com (Y.N.); 5Department of Neurosurgery, Kochi Health Sciences Center, Kochi 781-0111, Japan; yongran.12k04.ysw@gmail.com; 6Department of Neurosurgery, Hata Kenmin Hospital, Sukumo 788-0785, Japan; yokodani4543@gmail.com (M.Y.); msxtr3dodc@mvi.biglobe.ne.jp (N.M.); 7Department of Neurosurgery, Izumino Hospital, Kochi 781-0011, Japan; hoashiyu@gmail.com; 8Department of Neurosurgery, Mominoki Hospital, Kochi 780-0952, Japan; moriki1013@gmail.com; 9Department of Neurosurgery, Kochi Red Cross Hospital, Kochi 780-0026, Japan; kjbando0418@gmail.com (K.B.); nobuhisama@gmail.com (N.M.)

**Keywords:** aneurysmal subarachnoid hemorrhage, delayed admission, symptomatic vasospasm, functional outcomes, endovascular therapy, direct surgery

## Abstract

**Background/Objectives**: Aneurysmal subarachnoid hemorrhage (SAH) requires prompt treatment, yet hospital admission is occasionally delayed, and the optimal treatment strategy for such patients remains to be established. We aimed to investigate treatment modality, treatment timing, and outcomes in patients with SAH with respect to early versus delayed admission. **Methods**: A total of 1080 patients with SAH and a defined onset date were included in this prefecture-wide, multicenter, registry-based study. Baseline characteristics, late SAH complications (including vasospasm), and functional outcomes were compared between early and delayed admission groups at Day 4 or later (Day 0 = SAH onset). Additionally, the association of treatment choice (endovascular therapy or direct surgery) with treatment timing was analyzed in the delayed admission group. **Results**: Delayed admission was observed in 69 (6.4%) patients. The neurological status upon admission was significantly better in the delayed admission group, with more World Federation of Neurological Societies grades I–II (89.8% vs. 56.2% in the early admission group). Delayed admission was significantly associated with an increased incidence of symptomatic vasospasm by multivariable logistic regression analysis (odds ratio 2.51: 95% confidence interval 1.26–5.00, *p* = 0.009), while a significant difference in poor functional outcomes (modified Rankin scale 3–6) was not revealed. Although endovascular therapy use did not increase in the delayed admission group, the interval from admission to endovascular therapy was significantly shorter than that in the direct surgery group (0 [0–1] days vs. 1 [1–8] days: median [interquartile range], *p* = 0.007, Mann–Whitney U test). **Conclusions**: Delayed admission was a risk factor for symptomatic vasospasm; however, functional outcomes were not exacerbated. These results were obtained under the treatment strategy of multiple institutions, where the timing of endovascular therapy was earlier than that of direct surgery in patients with delayed admission.

## 1. Introduction

Aneurysmal subarachnoid hemorrhage (SAH) is a life-threatening condition associated with high mortality rates and the incidence of severe disability among survivors [1]. Early medical intervention (including treating the ruptured aneurysm) is recommended to prevent rebleeding, control intracranial pressure, and prevent delayed complications [2,3]. However, patients with acute SAH are not necessarily admitted to medical institutions immediately due to an underestimation of the SAH symptoms by the patients or via initial misdiagnosis at the first medical contact [4,5].

Previous studies have described the impact of delayed admission on SAH-related complications and functional outcomes primarily based on single-center experience, with mixed results depending on the definition of delayed admission and treatment strategy of the ruptured aneurysms [5,6,7]. In particular, when the patient is admitted during the period prone to delayed neurological deterioration due to vasospasm, the choice of an optimal treatment modality and its timing are still debatable to date.

In this article, we investigated the impact of delayed admission at Day 4 or later (Day 0 = SAH onset) on the incidence of symptomatic cerebral vasospasm and functional outcomes from a multicenter, prefecture-wide SAH registry in Japan. The timing of aneurysmal treatment and the choice of treatment modality were also analyzed and discussed in the delayed admission patient group.

## 2. Materials and Methods

The study was conducted based on the STROBE (Strengthening the Reporting of Observational Study in Epidemiology) criteria. The study protocol was approved by the Kochi Medical School Hospital Research Ethics Committee (Kochi, Japan) (#108921), and consent was waived because there were no unique patient identifiers. The institutional ethics committees of all participating hospitals also approved the study protocol.

### 2.1. Patient Selection, Clinical Evaluation, and Study Design

The Department of Health Policy in Kochi Prefectural Office, Japan, has registered all acute stroke patients who were admitted to 29 hospitals in Kochi prefecture since 2011 (Kochi Acute Stroke Survey of Onset: KATSUO registry). The characteristics of the stroke care system in Kochi Prefecture and the KATSUO registry database have been described previously [8,9]. Briefly, selection bias is minimized because all the prefecture-wide stroke patients are registered and because Kochi Prefecture is a ‘closed’ county regarding emergency patient transfer, where only about 1% of the patients are transferred into or out of Kochi Prefecture [9,10]. Data were provided for Kochi Medical School Hospital after patient anonymization. Between May 2011 and December 2022, 1451 patients with SAH were registered in the KATSUO registry from seven primary and comprehensive stroke centers in Kochi Prefecture. Among them, eight patients with an unknown onset date, 98 with non-aneurysmal SAH, and 53 with missing modified Rankin scale (mRS) scores at discharge were excluded. Among the remaining 1186 patients with aneurysmal SAH, 106 who did not undergo surgical or endovascular treatment were excluded from the study population. Consequently, 1080 patients who underwent treatment for the ruptured aneurysms were included in the final analysis. Figure 1 shows the flow diagram of the patient selection process.

This retrospective cohort study investigated the association between delayed hospital admission at Day 4 or later (Day 0 = SAH onset) and functional outcomes, symptomatic vasospasm, procedure-related complications, and chronic hydrocephalus in patients with SAH due to ruptured intracranial aneurysms. The threshold of Day 4 was determined on the basis that delayed cerebral ischemia due to vasospasm typically occurs after Day 3, and obliteration of the aneurysm within 72 h of onset has been reported to improve functional outcomes [11,12]. The date of SAH onset was deemed to be the time of occurrence of a headache that persisted, aggravated, recurred after initial improvement, and was either followed by a new neurological deficit or not. Symptomatic vasospasm was defined as a neurological worsening associated with significant narrowing of the major vessels by digital subtraction angiography, computed tomography (CT) angiography, or magnetic resonance angiography, after excluding other causes by laboratory and radiological examinations. The neurological worsening was characterized by a decrease in the Glasgow Coma Scale score of two points or greater or an increase in the motor score of the National Institutes of Health Stroke Scale of two points or greater, which lasted for more than 8 h [13]. Poor functional outcomes were defined as an increasing mRS score ≥ 3 at the time of discharge from the hospital for patients with a pre-admission mRS score of 0 or 1. Procedure-related complications were defined as neurological deterioration associated with cortical or subcortical cerebral infarction, postoperative hematoma formation, and increasing subarachnoid clot volume [14]. Chronic hydrocephalus was defined as symptomatic hydrocephalus requiring cerebrospinal fluid shunting. Age, sex, dates of SAH onset and hospital admission, and premorbid history (hypertension, smoking status) were retrieved from the KATSUO registry. In addition, the Fisher CT group, initial neurological status according to the World Federation of Neurosurgical Societies (WFNS) grade, treatment modality (direct surgery or endovascular therapy), the maximum diameter of the aneurysm, aneurysmal location (internal carotid artery [ICA], anterior cerebral artery [ACA], middle cerebral artery [MCA], and posterior circulation [PCQ]), symptomatic vasospasm, procedure-related complications, chronic hydrocephalus, and mRS score at the time of discharge were collected for the patient cohort from the participating hospitals. During the study period, the timing of therapeutic intervention, choice of treatment modality, and postoperative care were determined at the discretion of the participating hospitals and attending physicians.

Baseline characteristics of the delayed admission group (Day 4∼) were compared with those of the early admission group (∼Day 3). The association between delayed hospital admission and functional outcomes, symptomatic vasospasm, procedure-related complications, and chronic hydrocephalus was analyzed using univariate, multivariable, and subgroup analyses. Subsequently, the date of intervention and choice of treatment modality for the ruptured aneurysms in the delayed admission group were recorded in detail, along with the presence of angiographic vasospasm on admission, which was defined as ≥50% stenosis in the major cerebral arteries via magnetic resonance imaging, CT, or cerebral angiography.

### 2.2. Date Analysis

Quantitative variables were expressed as the median and interquartile range (IQR: 25th–75th percentile) as appropriate, and non-normally distributed continuous variables were analyzed using the Mann–Whitney U-test. Categorical variables were evaluated using the chi-square test. Univariate and multivariable logistic regression analyses were performed to determine the risk factors for symptomatic cerebral vasospasm and poor functional outcomes (mRS 3–6). The odds ratio (OR) and 95% confidence interval (CI) were also determined via logistic regression analyses. Multiple imputation was used to handle missing values. The number of multiple imputations was set at 10, and 50 iterations were performed. This created 10 complete datasets in which the missing values were imputed with different plausible values. Predictive mean matching was specified as the imputation method. Logistic regression analysis was performed on each imputed dataset, and the results were pooled to calculate the OR and 95% CI. SPSS version 29 (IBM Corp., Armonk, NY, USA) and R Statistical Software (v4.2.0; R Core Team 2022) were used for all statistical analyses. Probability (*p*) values < 0.05 were considered statistically significant, and all *p* values were two-sided.

## 3. Results

Patient baseline characteristics are summarized in Table 1. Delayed admission at Day 4 or later was observed in 69 (6.4%) cases, where 25 patients were delayed due to misdiagnosis at the first visit to medical institutions. All the patients experienced a headache with or without vomiting at the onset. Among them, 63 patients visited the hospital for persisting headache, two for drowsiness, one for general malaise, and one for gait disturbance. One patient exhibiting aphasia and the other displaying right leg weakness were considered to have symptomatic vasospasm upon admission.

Table 2 shows the results of a univariate analysis of factors for delayed admission. The location of the aneurysm was significantly different between the patient groups, where more ICA aneurysms and fewer MCA and PCQ aneurysms were observed in the delayed admission group. The neurological status upon admission, as represented by the WFNS grade, was significantly better in the delayed admission group, with more WFNS grades I–II (89.8% vs. 56.2%) and fewer WFNS grades IV–V (4.3% vs. 40.0%, respectively) cases compared with those in the early admission group. A patient with Fisher group 3, representing the presence of a thick subarachnoid clot in the head CT, was less likely to be observed in the delayed admission compared with the early admission group (44.9% vs. 83.8%, respectively).

Univariate analyses were used to assess an association between delayed patient admission and either treatment modalities or various outcome measures, and the results are shown in Table 3. The incidence of endovascular therapy use and the frequency of procedure-related complications were comparable with those of the early admission group. Symptomatic vasospasm was significantly more likely to be observed in the delayed versus the early admission group (20.3% vs. 11.9%, respectively; *p* = 0.042, chi-square test). Among 14 patients presenting symptomatic vasospasm in the delayed admission group, two, three, and nine patients had symptomatic vasospasm on admission, between admission and treatment, and after treatment, respectively. In the analysis for poor functional outcomes, 97 patients with premorbid mRS ≥ 2 were excluded so that the remaining 983 patients could be evaluated. Poor functional outcomes representing mRS scores of 3-6 at the time of discharge were less likely to be observed in the delayed compared with the early admission group (28.6% vs. 53.2%, respectively; *p* < 0.001, chi-square test) in the univariate analysis.

An association between delayed admission and symptomatic vasospasm was further evaluated by multivariable logistic regression analyses. After adjustment for other covariates such as WFNS grade and endovascular therapy use, delayed admission was observed to be an independent risk factor for symptomatic vasospasm (OR = 2.51, 95% CI [1.26–5.00], *p* = 0.009) (Table 4). Motivated by the significant differences in WFNS grade between the early and delayed admission groups, we undertook subgroup analysis and multivariable logistic regression analyses to investigate the association between delayed admission and poor functional outcomes. The subgroup analysis of 627 patients with WFNS grades I–III revealed 17/60 (28.3%) and 207/567 (36.5%) patients with poor functional outcomes in the early and delayed admission groups, respectively.

The association between delayed admission and poor functional outcomes was not revealed in a statistically significant fashion by multivariable analysis (OR = 0.53, 95% CI [0.28–1.02], *p* = 0.059) (Table 5). The choice of treatment modality in the delayed admission group and its association with the timing of aneurysmal treatment, angiographical vasospasm upon admission, procedure-related complications, functional outcomes, and location of the aneurysm were further analyzed (Table 6). Comparing the endovascular therapy with the direct surgery group, the interval from admission to treatment was significantly shorter (0 [0–1] vs. 1 [1–8] days, median [IQR], respectively; *p* = 0.007, Mann–Whitney U test), and the ratio of treatment within 4 days after admission was significantly higher (91.9% vs. 68.8%, respectively; *p* = 0.028, Mann–Whitney U test). None of the patients experienced rebleeding between admission and treatment. Procedure-related complications (composite of ischemic and hemorrhagic) occurred in 7/37 (18.9%) and 6/32 (18.8%) patients of the endovascular therapy and direct surgery groups, respectively. The incidence of poor functional outcomes was also comparable between these two groups.

Patients with angiographic vasospasm upon admission were treated by endovascular therapy (eight cases), all of whom were treated before the angiographic vasospasm resolved. Direct surgery was performed in six patients with angiographic vasospasm upon admission before (three cases) and after (three cases) resolution of the vasospasm. More PCQ aneurysms and fewer MCA aneurysms were treated by endovascular therapy than by direct surgery in the delayed admission group, and this statistical trend was similar to that observed in the early admission group.

## 4. Discussion

In this study, we investigated the association of delayed admission at Day 4 or later with complications and outcomes of aneurysmal SAH from a multicenter, prefecture-wide registry in Japan. Delayed admission was found to be an independent risk factor for symptomatic vasospasm. Although more patients with better functional outcomes were observed in the delayed admission group, this positive impact of delayed admission did not achieve statistical significance after adjustment for potential confounders. We also described interactions among choice of treatment modality, timing of treatment, and procedure-related complications of the ruptured aneurysms in the delayed admission group of patients.

Aneurysmal SAH remains a life-threatening condition associated with high mortality and morbidity [1]. Immediate occlusion of the culprit aneurysm and subsequent neurocritical care are recommended to prevent secondary brain injury caused by rebleeding and vasospasm [15]. However, hospital admission of patients with SAH may sometimes be delayed for several reasons, including misdiagnosis at the initial contact with the medical service as well as delay in seeking medical help by the patients themselves, who do not consider their symptoms serious [5]. Previous studies have investigated the prognosis of SAH with delayed admission, resulting in controversy presumably due to a lack of consensus for the definition of delayed admission [5,6,7,16,17]. When studies exclusively focus on patients who were misdiagnosed by clinicians, significantly worse outcomes were reported in the group of patients with delayed admission [6,16]. However, when a study also included a delay in seeking medical help on the part of the patient, which is more prevalent than misdiagnosis in the current era with more accurate and widespread neurovascular imaging, better functional outcomes were observed in patients with delayed admission [5,7]. This discrepancy in functional outcome results may have been caused by a better initial neurological status (e.g., WFNS grades I-–I) for patients reporting a delayed visit to the hospital. A study including patients with a delayed hospital visit does not capture patients presenting with severe neurological deficit upon admission, who cannot describe their first ictus, resulting in a selection bias. Also, the initial symptoms of those patients who sought medical attention at the onset and were misdiagnosed should be more severe than those of patients who took a wait-and-see approach and chose to stay at home. Although patients who survived for several days after the first ictus without rebleeding [18,19] generally exhibit better functional outcomes that are associated with a better neurological status upon admission, we have been wondering whether the outcomes of patients with delayed admission are different from those of the early admission group after adjustment for confounding factors such as initial WFNS grade.

In this study, delayed hospital admission was found to be an independent risk factor for symptomatic cerebral vasospasm. There are several possible reasons for this finding. A subarachnoid clot is not effectively cleaned out early after the onset in the delayed admission group. The remaining subarachnoid clot contains vasoconstrictive factors, including products of the vascular endothelium and erythrocyte degradation [20,21]. An early washout of the subarachnoid clot by surgical removal, cerebrospinal fluid drainage, and spontaneous mechanisms has been reported to decrease the risk of vasospasm [22,23,24]. Furthermore, early anti-vasospastic treatment is available, particularly after the aneurysmal repair [25]. Fasudil hydrochloride and ozagrel sodium, which are approved as anti-vasospastic agents in Japan, were used in this study period, but administration of such agents was missing or delayed in the delayed admission group [26]. Notably, a delayed hospital admission has also been shown to be significantly associated with symptomatic vasospasm in a previous study by Goertz et al., which has a different study design that employs a definition of delayed admission as ≥48 h and involves a more predominant use of endovascular therapy [7], which may explain the robustness of the findings in our study.

In this study, we did not find significantly different functional outcomes between the early and delayed admission groups after adjustment for confounding factors by multivariable analysis, suggesting that the seemingly more favorable outcomes were attributable to a better neurological status upon admission in the delayed admission group. Considering the better neurological status on admission, comparable functional outcomes might suggest relative deterioration of the patients with delayed admission due to the increased incidence of vasospasm. Several controversies exist in the treatment modality and treatment timing for ruptured aneurysms of the patients in the delayed admission group. More frequent use of less invasive endovascular therapy may be advantageous when dealing with brain tissues and vessels showing signs of vasospasm, while an overuse of endovascular therapy may increase procedure-related complications of the aneurysm that is amenable to direct surgery [7,27]. Additionally, although an immediate obliteration of the ruptured aneurysm offers maximal protection from rebleeding and provides anti-vasospastic treatment, delaying aneurysmal obliteration for >10 days after SAH onset may minimize the risk of procedure-related complications by avoiding insult during the unstable hemodynamic conditions prevalent in the vasospastic period [17]. In our present study, attending physicians in multiple participating hospitals, on average, did not increase the use of endovascular therapy but preferred it when immediately treating the aneurysm in the delayed admission group. A comparable incidence rate of procedure-related complications between early endovascular therapy performed during the vasospastic period and late direct surgery performed during the post-vasospastic period may suggest the validity of endovascular therapy as an immediate intervention for patients who were admitted in the vasospastic period. We believe that the treatment strategy for ruptured aneurysms in the delayed admission group should be individualized according to multiple factors, including patient characteristics, timing of hospital visit, anatomical conditions of the aneurysm, and the presence of an angiographic vasospasm. For example, the clinical management responses to a patient admitted on Day 4 with no signs of vasospasm and the one who is admitted on Day 9 with a severe angiographic vasospasm may be varied. Also, even though an elective direct surgery may have been planned after the vasospasm subsides, a drastic change in the shape of the aneurysm revealed by neuroimaging may urge the attending physician to treat the aneurysm immediately. In our study, none of the patients in the delayed admission group experienced rebleeding before treatment, presumably due to the relatively low daily rebleeding rate at Day 4 or later in our study [18,19], which may justify a treatment strategy where attending physicians wait for several days until the optimal timing of intervention.

This study has several limitations. First, our findings should be interpreted with caution because of the observational and retrospective study design. Second, because of the small number of the patients, an association of functional outcomes with patients’ baseline characteristics, treatment modality, treatment timing, presence of angiographic or symptomatic vasospasm, anatomical condition of the aneurysm, and their interactions was only descriptive and thus less conclusive in the delayed admission group. Given this study was observational and descriptive in nature, the findings do not necessarily lead to immediate actionable consequences for clinical practice. Third, a direct comparison between early and delayed admission groups is intrinsically challenging even with multivariable or subgroup analysis. For example, a comparison of clot volume at presentation may be less relevant, where the patients with a Fisher group 2 subarachnoid clot in the delayed admission group may have had a Fisher group 3 SAH at onset before washout. Fourth, longer-term outcomes after discharge are unavailable, which limits the validity of conclusions about functional recovery. Further investigations, using a nationwide registry with more patients, for example, are warranted to identify individualized optimal treatment strategies in this heterogeneous patient subgroup.

## 5. Conclusions

In this study, using a prefecture-wide multicenter registry in Japan, a delayed admission at Day 4 or later was observed to be significantly associated with an increased incidence of symptomatic vasospasm, while functional outcomes remained comparable to those of the early admission group. The use of endovascular therapy did not increase in the delayed admission group, but it was used significantly earlier than in the direct surgery.

## Figures and Tables

**Figure 1 jcm-14-03537-f001:**
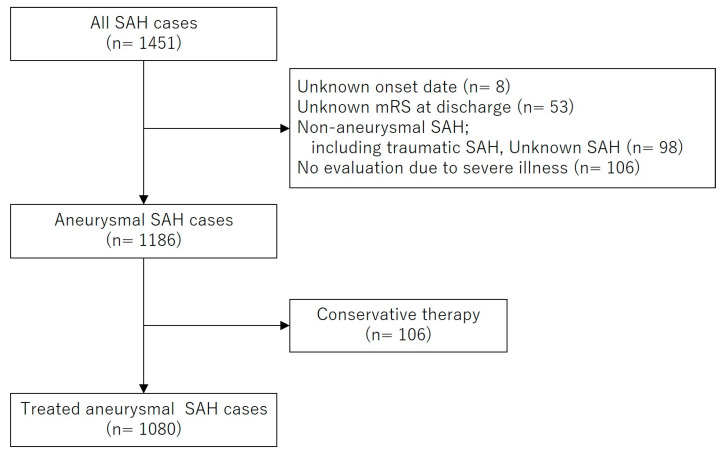
Flow diagram of the patient selection process.

**Table 1 jcm-14-03537-t001:** Baseline characteristics of 1080 patients with SAH.

Variable	Value	Missing Value
Mean age (years)	67 [55–78]	
Woman	778 (72.0)	
Hypertension	620 (57.4)	41 (3.8)
Smoking	369 (34.2)	65 (6.0)
Premorbid mRS 0 or 1	983 (91.0)	2 (0.2)
Delayed admission	69 (6.4)	
Location of aneurysm		
ICA	340 (31.5)	
ACA	363 (33.6)	
MCA	219 (20.3)	
PCQ	158 (14.6)	
Aneurysm size in mm	5.4 [3.9–7.5]	53 (4.9)
WFNS grade		5 (0.5)
I	362 (33.5)	
II	268 (24.8)	
III	38 (3.5)	
IV	189 (17.5)	
V	218 (20.2)	
Fisher group		3 (0.3)
1	14 (10.6)	
2	126 (11.7)	
3	878 (81.5)	
4	59 (5.6)	
Endovascular therapy	533 (49.4)	
Procedure-related complications	230 (21.3)	8 (0.7)
Ischemic	204 (19.0)	
Hemorrhagic	40 (3.2)	
Symptomatic vasospasm	134 (12.5)	5 (0.5)
Chronic hydrocephalus	239 (22.1)	4 (0.4)
Poor functional outcomes (mRS 3-6)	593 (54.9)	

Data are presented as the number of patients (%) or median [interquartile range]. ICA = internal carotid artery; ACA = anterior cerebral artery; MCA = middle cerebral artery; PCQ = posterior circulation; WFNS = World Federation of Neurosurgical Societies; mRS = modified Rankin scale.

**Table 2 jcm-14-03537-t002:** Univariate analysis of factors involved in delayed admission.

Variable	Early Admission	Delayed Admission	*p* Value
Age	66 [55–78]	71 [58–78]	0.19
Woman	731 (72.3)	47 (68.1)	0.45
Hypertension	583 (57.7)	37 (53.6)	0.54
Smoking	348 (34.4)	21 (30.4)	0.60
Premorbid mRS 0 or 1	920 (90.1)	63 (94.0)	0.40
Location of aneurysm			**0.041**
ICA	310 (30.7)	30 (43.5)	
ACA	338 (33.4)	25 (36.2)	
MCA	212 (21.0)	7 (10.1)	
PCQ	151 (14.9)	7 (10.1)	
Aneurysm size	5.4 [4.0–7.5]	5.8 [3.8–7.0]	0.80
WFNS grade			**<0.** **001**
I	311 (30.8)	51 (73.9)	
II	257 (25.4)	11 (15.9)	
III	36 (3.6)	2 (2.9)	
IV	187 (18.5)	2 (2.9)	
V	217 (21.5)	1 (1.4)	
Fisher group			**<0.001**
1	6 (0.6)	8 (11.6)	
2	98 (9.7)	28 (40.6)	
3	847 (83.8)	31 (44.9)	
4	57 (5.6)	2 (2.9)	

Data are presented as the number of patients (%) or median [interquartile range]. Categorical variables are analyzed by the chi-square test. Continuous variables are analyzed by the Mann–Whitney U test. Statistically significant variables are indicated by boldface. ICA = internal carotid artery; ACA = anterior cerebral artery; MCA = middle cerebral artery; PCQ = posterior circulation; WFNS = World Federation of Neurosurgical Societies; mRS = modified Rankin scale.

**Table 3 jcm-14-03537-t003:** Univariate analysis of the association of delayed admission with treatment modalities and various outcome measures.

Variable	Early Admission *n* = 1011	Delayed Admission *n* = 69	*p* Value
Endovascular therapy	496 (49.1)	37 (53.6)	0.46
Procedure-related complications	217 (21.6)	13 (18.8)	0.58
Symptomatic vasospasm	120 (11.9)	14 (20.3)	**0.04** **2**
Poor functional outcomes (mRS 3-6)	489 (53.2)	18 (28.6)	**<0.001**

Data are presented as the number of patients (%). Categorical variables are analyzed by chi-square test. Patients with premorbid mRS ≥ 2 (*n* = 95) are excluded from the analysis of poor functional outcomes. Statistically significant variables are indicated by boldface. mRS = modified Rankin scale.

**Table 4 jcm-14-03537-t004:** Results of univariate and multivariable logistic regression analysis of risk factors for symptomatic vasospasm.

Variable	Symptomatic Vasospasm *n* = 134	No symptomatic Vasospasm *n* = 941	Univariate Analysis OR (95% CI)	*p* Value	Multivariable Analysis OR (95% CI)	*p* Value
Age	66 [55–77]	68 [58–78]	1.01 (0.99–1.03)	0.21	1.01 (0.99–1.03)	0.29
Woman	92 (68.7)	681 (72.4)	0.86 (0.57–1.24)	0.37	0.68 (0.42–1.11)	0.13
Hypertension	79 (59.0)	540 (57.4)	1.07 (0.73–1.56)	0.73	1.01 (0.67–1.51)	0.97
Smoking	43 (32.1)	325 (34.5)	0.89 (0.60–1.32)	0.57	0.79 (0.47–1.33)	0.37
Location of aneurysm						
ICA	41 (30.6)	296 (31.5)	Ref		Ref	
ACA	50 (37.3)	312 (33.2)	1.16 (0.74–1.80)	0.52	1.09 (0.68–1.76)	0.71
MCA	32 (23.9)	186 (19.8)	1.24 (0.76–20.4)	0.39	0.85 (0.50–1.45)	0.56
PCQ	11 (8.2)	147 (15.6)	0.54 (0.27–1.08)	0.082	0.82 (0.39–1.71)	0.59
Aneurysm size	5.4 [3.9–7.5]	5.6 [4.2–7.5]	1.00 (0.95–1.06)	0.91	0.99 (0.94–1.05)	0.89
WFNS grades IV–V	58 (43.3)	345 (36.8)	1.30 (0.90–1.88)	0.16	1.57 (1.05–2.33)	**0.027**
Fisher group 3	110 (82.1)	764 (81.2)	1.04 (0.65–1.67)	0.86	1.38 (0.82–2.30)	0.23
Delayed admission	14 (10.4)	55 (5.8)	1.88 (1.01–3.48)	**0.045**	2.51 (1.26–5.00)	**0.009**
Endovascular therapy	38 (28.4)	493 (52.4)	0.36 (0.24–0.54)	**<0.001**	0.32 (0.21–0.50)	**<0.001**

Data are presented as the number of patients (%) or median [interquartile range]. Statistically significant variables are indicated by boldface. ICA = internal carotid artery; ACA = anterior cerebral artery; MCA = middle cerebral artery; PCQ = posterior circulation; WFNS = World Federation of Neurosurgical Societies.

**Table 5 jcm-14-03537-t005:** Results of univariate and multivariable logistic regression analysis of risk factors for poor functional outcomes (mRS ≥ 3) at discharge.

	Favorable Outcomes *n* = 476	Poor Outcomes *n* = 507	Univariate Analysis OR (95% CI)	*p* Value	Multivariable Analysis OR (95% CI)	*p* Value
Age	60 [49–68.75]	70.0 [59–79]	1.05 (1.04–1.06)	**<0.001**	1.06 (1.04–1.08)	**<0.001**
Female sex	316 (66.4)	376 (74.2)	1.45 (1.10–1.91)	**0.008**	0.97 (0.66–1.45)	0.90
Hypertension	242 (50.8)	313 (61.7)	1.66 (1.28–2.16)	**<0.001**	1.28 (0.92–1.77)	0.14
Smoking	211 (44.3)	148 (29.2)	0.51 (0.39–0.67)	**<0.001**	0.67 (0.45–0.98)	**0.04**
Location of aneurysm						
ICA	153 (32.1)	145 (28.6)	Ref		Ref	
ACA	165 (34.7)	175 (34.5)	1.12 (0.82–1.53)	0.48	1.42 (0.96–2.11)	0.08
MCA	93 (19.5)	108 (21.3)	1.23 (0.86–1.75)	0.27	1.10 (0.69–1.76)	0.69
PCQ	65 (13.7)	79 (15.6)	1.28 (0.86–1.91)	0.22	1.33 (0.80–2.20)	0.28
Aneurysm size	5.0 [3.7–6.8]	5.8 [4.1–8.2]	1.09 (1.05–1.14)	**<0.001**	1.08 (1.03–1.13)	**0.002**
WFNS grade 4 or 5	71 (14.9)	282 (55.6)	7.15 (5.26–9.72)	**<0.001**	8.43 (5.88–12.1)	**<0.001**
Fisher group 3	368 (77.3)	439 (86.6)	1.98 (1.41–2.78)	**<0.001**	1.58 (1.03–2.43)	**0.038**
Delayed admission	45 (9.5)	18 (3.6)	0.35 (0.20–0.62)	**<0.001**	0.53 (0.28–1.02)	0.059
Endovascular therapy	215 (45.2)	259 (51.1)	1.27 (0.99–1.63)	0.06	1.01 (0.71–1.42)	0.96

Data are presented as the number of patients (%) or median [interquartile range]. Statistically significant variables are indicated by boldface. ICA = internal carotid artery; ACA = anterior cerebral artery; MCA = middle cerebral artery; PCQ = posterior circulation; WFNS = World Federation of Neurosurgical Societies; mRS = modified Rankin scale.

**Table 6 jcm-14-03537-t006:** Choice of treatment modality and its association with the timing of aneurysmal treatment, angiographical vasospasm on admission, and location of the aneurysm in the delayed admission group.

	Endovascular Therapy *n* = 37	Direct Surgery *n* = 32	*p* Value
Interval from admission to treatment	0 [0–1]	1 [1–8]	**0.007**
Treatment within 4 days after admission	34 (91.9)	22 (68.8)	**0.028**
Procedure-related complications	7 (18.9)	6 (18.8)	0.99
Poor functional outcomes	14 (37.8)	8 (25.0)	0.38
Angiographical vasospasm on admission	8 (21.6)	6 (18.8)	0.77
Location of aneurysm			
ICA	17 (45.9)	12 (37.5)	
ACA	11 (29.7)	14 (43.8)	
MCA	2 (5.4)	6 (18.8)	
PCQ	7 (18.9)	0 (0)	

Data are presented as the number of patients (%) or median [interquartile range]. Statistically significant variables are indicated by boldface. ICA = internal carotid artery; ACA = anterior cerebral artery; MCA = middle cerebral artery; PCQ = posterior circulation.

## Data Availability

The raw data supporting the conclusions of this article will be made available by the authors upon request.

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
