# Peer review of "Impact of Delayed Admission on Treatment Modality and Outcomes of Aneurysmal Subarachnoid Hemorrhage: A Prefecture-Wide, Multicenter Japanese Study"

_jcm, 2025, doi:10.3390/jcm14103537_

Round 1

Reviewer 1 Report

Comments and Suggestions for Authors

Summary:
The present study analyzes the impact of delayed hospital admission on treatmnt modality and outcomes in patients with aneurysmal subarachnoid hemorrhage (SAH) using data from a large, prefecturewide multicenter registry. The focus is particullarly on the incidence of symptomatic vasospasm and functional outcome at discharge. Overall, the manuscript is carefully prepared, methodologically sound, and free from apparent errors. It is evident that the authors worked in a detailed and thorough manner throughout. No carelessness or inconsistencies were noted and I am largely satisfied with the manuscript.

Points for Improvement in my opptinion:

  1. Definition of Symptomatic Vasospasm (Page 3, Line 98 and following):
    The definition of symptomatic vasospasm appears to reflect the criteria for delayed cerebral ischemia (DCI), describing neurological deterioration without identifiable causes. However, there is no explicit mention that concurrent vasospasm was confirmed by imaging ( was CTA, DSA, or similar performed?). Since DCI can occur without a visible macrovasospasm, the current definition seems too vague in my oppinion. Please clarify whether imaging confirmation was required for the diagnosis of symptomatic vasospasm.
  2. Discussion on Intracranial Pressure (Page 9, Line 293):
    The manuscript suggests that elevated intracranial pressure may contribute to symptomatic vasospasm in delayed admissions. However, in most of the patients who present several days after onset with most of the time relatively mild symptoms, elevated intracranial pressure is unlikely. This needs to be reconsidered or discussed in a more differentiated manner.
  3. Mechanisms Leading to Increased Vasospasm (Page 9, Line 296):
    In my opinion, the most important factor ist the missing antivasospastic therapie the patients do not receive, if they present late to. (e.g., timely administration of nimodipine or other preventive measures). This point deserves more emphasis / should be stated more clear
  4. A major limitation that should be explicitly addressed is that functional outcomes were only assessed at hospital discharge as far as i understood- There is no information on longer-term outcomes (e.g., 3–6 months), limiting the validity of conclusions about functional recovery. This point should be clearly discussed among the study limitations.
  5. The study remains largely observational and descriptive in nature. While the findings are interesting, they do not lead to immediate actionable consequences for clinical practice. This point could be acknowledged briefly in the discussion.

Minor Points:

  1. The discussion section is relatively extensive and could be more concise without losing critical content.

Author Response

1. Definition of Symptomatic Vasospasm (Page 3, Line 98 and following):
The definition of symptomatic vasospasm appears to reflect the criteria for delayed cerebral ischemia (DCI), describing neurological deterioration without identifiable causes. However, there is no explicit mention that concurrent vasospasm was confirmed by imaging ( was CTA, DSA, or similar performed?). Since DCI can occur without a visible macrovasospasm, the current definition seems too vague in my oppinion. Please clarify whether imaging confirmation was required for the diagnosis of symptomatic vasospasm.

Response: The definition of symptomatic vasospasm in our registry includes significant narrowing of the major vessels by digital subtraction angiography, CT angiography, of magnetic resonance angiography, but I found I did not describe it in the manuscript. I added “significant narrowing of the major vessels by digital subtraction angiography, CT angiography, of magnetic resonance angiography” to the definition of symptomatic vasospasm in Page 3, Line 99-102.

2. Discussion on Intracranial Pressure (Page 9, Line 293):
The manuscript suggests that elevated intracranial pressure may contribute to symptomatic vasospasm in delayed admissions. However, in most of the patients who present several days after onset with most of the time relatively mild symptoms, elevated intracranial pressure is unlikely. This needs to be reconsidered or discussed in a more differentiated manner.

Response: I agree that intracranial pressure does not elevate so much in patients with delayed admission, who generally have mild symptoms. We deleted these sentences, and discuss delayed or missing anti-vasospastic agents in depth as a major contributor of increased incidence of symptomatic vasospasm (Page 9, Line 301-307).

3. Mechanisms Leading to Increased Vasospasm (Page 9, Line 296):
In my opinion, the most important factor ist the missing antivasospastic therapie the patients do not receive, if they present late to. (e.g., timely administration of nimodipine or other preventive measures). This point deserves more emphasis / should be stated more clear

Response: I agree that we should discuss missing anti-vasospastic therapy more as a major contributor of symptomatic vasospasm in the patients with delayed admission. Although nimodipine is unavailable, we used fasudil hydrochloride and ozagrel sodium, which are approved as an anti-vasospastic agent in Japan. I added such statements in (Page 9, Line 305-307), and change the reference article accordingly (Page 13, Line 456).

4. A major limitation that should be explicitly addressed is that functional outcomes were only assessed at hospital discharge as far as i understood- There is no information on longer-term outcomes (e.g., 3–6 months), limiting the validity of conclusions about functional recovery. This point should be clearly discussed among the study limitations.

Response: I agree that lack of information about long-term outcomes limits the validity of conclusions about functional recovery. I added it in the study limitation paragraph (Page 11, Line 365-366).

5. The study remains largely observational and descriptive in nature. While the findings are interesting, they do not lead to immediate actionable consequences for clinical practice. This point could be acknowledged briefly in the discussion.

Response: Acknowledging that the findings in this study do not necessarily lead to immediate actionable consequences is important. I added it in the study limitation paragraph (Page 10, Line 358- Page 11, Line 360).

Minor Points:

  1. The discussion section is relatively extensive and could be more concise without losing critical content.I agree. I deleted redundant parts of discussion without losing critical content (Page 9, Line 261-264), (Page 9, Line 276-278), (Page 9, Line 290-293), (Page 10, Line 316-318), (Page 10, Line 346-352).

Reviewer 2 Report

Comments and Suggestions for Authors

   This paper presents a retrospective analysis of subarachnoid hemorrhage (SAH) patients with delayed hospital admission, discussing their treatment and outcomes. The study addresses an important clinical issue, as delayed presentation may significantly impact prognosis. The findings could provide valuable insights into the management of such cases. However, several key questions need to be clarified to strengthen the manuscript.

he study focuses on delayed admission but does not clearly explain the primary reasons for the delay ,What was the misdiagnosis rate in this cohort?

Re-rupture is a critical concern in SAH, particularly in untreated aneurysms.How many patients in the delayed group experienced re-bleeding before treatment?

The paper mentions vasospasm but does not analyze whether delayed admission increases vasospasm risk.Was there a correlation between admission delay and the severity/timing of vasospasm?

Author Response

he study focuses on delayed admission but does not clearly explain the primary reasons for the delay ,What was the misdiagnosis rate in this cohort?

Response: Among 69 patients with delayed admission, misdiagnosis at the first contact to the medical institution was observed in 25 patients. We added this sentence at Page 4, Line 149-150.

Re-rupture is a critical concern in SAH, particularly in untreated aneurysms.How many patients in the delayed group experienced re-bleeding before treatment?   Response: None of the patients in the delayed group experienced re-bleeding between admission and treatment. It is probably because re-bleeding rate decreases after Day 3 and most of the patients in the delayed group were treated relatively early (within 4 days from admission). This may suggest justify treatment strategy where attending physicians wait for several days until the optimal timing of intervention. We added such descriptions at Page 7, Line 228-229 and Page 10, Line 343-346.   The paper mentions vasospasm but does not analyze whether delayed admission increases vasospasm risk.Was there a correlation between admission delay and the severity/timing of vasospasm ?   Response: Although correlation between admission delay and severity/timing of vasospasm is an interest of the readers, we do not have such data in our database. We only have information about timing of symptomatic vasospasm in the delayed admission group. Among 14 patients who had symptomatic vasospasm in the delayed admission group, two had symptomatic vasospasm on admission. Three experienced symptomatic vasospasm between admission and treatment, and nine had one after the treatment. This information is also important so we added it at Page 6, Line 183-185.
